# N-Terminal of the Prohormone Brain Natriuretic Peptide Predicts Postoperative Cardiogenic Shock Requiring Extracorporeal Membrane Oxygenation

**DOI:** 10.3390/jcm11195493

**Published:** 2022-09-20

**Authors:** Piotr Duchnowski

**Affiliations:** Cardinal Wyszynski National Institute of Cardiology, Alpejska 42, 04-628 Warsaw, Poland; pduchnowski@ikard.pl

**Keywords:** NT-proBNP, cardiogenic shock, valve surgery, extracorporeal membrane oxygenation (ECMO)

## Abstract

Aims: Heart valve surgery is associated with a risk of serious postoperative complications including postoperative cardiogenic shock (described as postcardiotomy shock (PCS)). The indication for extracorporeal membrane oxygenation (ECMO) is cardiogenic shock, which is resistant to optimal causal and pharmacological treatment, including the supply of catecholamines and/or an intra-aortic balloon pump (IABP). The aim of this study was to assess the usefulness of the selected preoperative biomarkers in the prediction of postoperative cardiogenic shock requiring ECMO in patients undergoing heart valve surgery. Methods: A prospective study was conducted on a group of consecutive patients with significant valvular heart disease that underwent elective valve surgery. The primary endpoint at the intra-hospital follow-up was postoperative cardiogenic shock requiring ECMO. Univariate analysis, followed by multivariate regression analysis, were performed. Results: The study included 610 patients. The primary endpoint occurred in 15 patients. At multivariate analysis, the preoperative N-terminal of the prohormone brain natriuretic peptide (NT-proBNP) level (OR 1.022; 95% CI 1.011–1.034; *p* = 0.001) remained an independent predictor of the primary endpoint. Conclusions: An elevated NT-proBNP level was associated with a higher risk of postoperative cardiogenic shock requiring the use of ECMO.

## 1. Introduction

Heart valve surgery is associated with a risk of serious postoperative complications including postoperative cardiogenic shock (described as postcardiotomy shock (PCS)) [1,2,3,4]. PCS is poorly defined in the literature, but it is broadly understood to mean circulatory failure after cardiac surgery which is resistant to inotropic support and/or an intra-aortic balloon pump (IABP), and which requires mechanical circulatory support (MCS) such extracorporeal membrane oxygenation (ECMO). ECMO may improve a patient’s physiological state when in cardiogenic shock by stabilizing hemodynamics and tissue metabolism, allowing the necessary time for regeneration of the heart muscle [5,6,7,8]. The available literature has little information on postcardiotomy shock in patients undergoing heart valve surgery. Previous studies have indicated that surgery on cardiac arrest, reduced preoperative left ventricular systolic function, or high-sensitivity troponin T (hs-TnT) measured immediately after surgery are predictors of postoperative cardiogenic shock [9,10]. The N-terminal of the prohormone brain natriuretic peptide (NT-proBNP) is a prohormone secreted into the blood mainly by left ventricular cardiomyocytes, which participates in the maintenance of cardiovascular homeostasis. The active form of the hormone—brain natriuretic peptide (BNP)—causes an increase in glomerular filtration, a decrease in sodium reabsorption in the kidney, the inhibition of aldosterone secretion, and a decrease in the activity of the sympathetic system. The process of NT-proBNP secretion by cardiomyocytes occurs in response to an increase in their voltage in the course of increased pre- and/or after-load. An increase in the concentration of BNP and the prohormone NT-proBNP in the blood is demonstrated, among other things, by the activation of the compensation mechanism, which already occurs in the period preceding the appearance of symptoms of heart failure [11,12,13,14]. The predictive ability of NT-proBNP has been reported in numerous publications on various cardiovascular disorders, including aortic stenosis, heart failure, coronary artery disease, myocardial infarction, congenital heart disease, and postoperative hemodynamic instability [15,16,17,18,19,20,21]. However, there is no information regarding NT-proBNP as a predictor of persisting cardiogenic shock despite intensive conservative treatment including the use of catecholamines and/or an IABP. Therefore, in the presented study, we attempted to assess the usefulness of NT-proBNP levels in predicting cardiogenic shock requiring ECMO.

## 2. Methods

This is a prospective study of consecutive patients with hemodynamically significant valvular heart disease (aortic stenosis, aortic regurgitation, mitral stenosis, or mitral regurgitation) that were approved for heart valve surgery and subsequently underwent elective replacement or repair of the valve/valves, with or without a concomitant coronary artery bypass graft (CABG) at the National Institute of Cardiology, Warsaw, Poland. The exclusion criteria were a lack of consent to participate in the study, patients under 18 years of age, porcelain aorta, active infective endocarditis, and active neoplastic diseases. The day before surgery, a blood sample was collected from each patient. The plasma levels of the NT-proBNP concentrations were measured by electrochemiluminescent immunoassays using Elecsys 2010 (Roche, Mannheim, Germany) and the plasma levels of the cardiac TnT (cTnT) concentrations were measured by the troponin T hs-STAT (Roche, Mannheim, Germany). All procedures were performed through a midline sternotomy incision under general anaesthesia in a normothermia state. All patients were given cold blood cardioplegia at an initial dose of 15–20 mL/kg, followed by booster doses of 5–10 mL/kg every 20 min. The primary endpoint at intra-hospital follow-up was postoperative cardiogenic shock requiring the use of ECMO. Postoperative cardiogenic shock was diagnosed in patients with a systolic blood pressure of below 90 mm Hg and symptoms of organ hypoperfusion (cold viscous skin, altered mental state, oliguria, and increased serum lactate level) that were resistant to inotropic support and/or an IABP. The decision to use the arterio-venous ECMO was made by the team of physicians responsible for the patient who was observed to continue cardiogenic shock despite intensive conservative treatment, including the use of catecholamines and possibly an IABP. Patients were observed until discharge from the hospital or until death. The protocol was approved by the Institutional Ethics Committee of the Institute of Cardiology, Warsaw, Poland (number 1705).

## 3. Statistical Analysis

Statistical analysis was performed with IBM SPSS software, version 2.0 (SPSS Inc., Chicago, IL, USA). Data are presented as medians with ranges if continuous, or as frequencies if categorical. Binary logistic regression was used to assess the relationships between the variables. The following preoperative covariates were investigated for association with the primary endpoint in univariate analysis: age, BMI, bilirubine, creatinine, hs-TnT, NT-proBNP, haemoglobin level, red cell distribution width, left ventricular ejection fraction (LVEF), New York Heart Association (NYHA) classes, heart rate before surgery, blood pressure before surgery, tricuspid annular plane systolic excursion (TAPSE), right ventricular systolic pressure (RVSP), atrial fibrillation, moreover aortic cross-clamp time, cardiopulmonary bypass time, aortic valve plasty (AVP), aortic valve replacement (AVR), CABG, mitral valve plasty (MVP), mitral valve replacement (MVR), AVR plus MVR and major blending after surgery for the entire patient group, and patient subgroups with and without atrial fibrillation. Significant determinants (*p* < 0.05) identified from the univariate analysis for the entire patient group were subsequently entered into multivariate models. The predictive values of NT-proBNP were assessed by a comparison of the areas under the receiver operator characteristics of the respective curve. On the basis of the Youden index, a cut-off point was determined that met with the criterion of maximum sensitivity and specificity for mortality prediction. Spearman’s rank correlation test was performed to examine possible associations between the variables describing the severity of heart failure and myocardial damage, that is, the test for associations between the values of NT-proBNP and NYHA classes, LVEF, TAPSE, and hs-TnT.

## 4. Results

The present study included 610 patients undergoing heart valve surgery, with or without concomitant procedures on coronary arteries. The mean age in the studied population was 63 (±12). Forty-nine (8%) patients had a significantly impaired LVEF (LVEF ≤ 35%) before cardiac surgery. The mean preoperative NT-proBNP level was 2003 pg/mL (standard deviation (SD) ± 1532). Table 1 presents the characteristics of the entire study group. Postoperative cardiogenic shock requiring ECMO occurred in 15 patients. In eight cases, an IABP was used (these were patients with haemodynamic instability in the immediate postoperative period that did not respond to increased doses of catecholamines before leaving the operating table). Due to the lack of stabilization of hemodynamic parameters, the IABP was replaced with ECMO in five cases. The average ECMO implantation time after the surgery was completed was 18 h. In each case, the indication for ECMO implantation was increasing hemodynamic instability accompanied by an increase in tissue hypoxia markers. The statistically significant predictors of the primary endpoint at univariate analysis are presented in Table 2 (univariate analysis showed a trend towards statistical significance of the cardiopulmonary bypass time parameter for the primary endpoint, *p* = 0.07). At multivariate analysis, only NT-proBNP (OR 1.022; 95% CI 1.011–1.034; *p* = 0.001) remained an independent predictor of the primary endpoint. The area under the receiver operator characteristic curve for the primary endpoint for NT-proBNP was 0.764 (95% CI 0.728–0.797) (sensitivity: 67%; specificity: 79%) (Figure 1). The mean preoperative value of NT-proBNP in patients with postoperative cardiogenic shock requiring ECMO was 7053 pg/mL (±3532) and was significantly higher compared to patients with no postoperative cardiogenic shock 1875 pg/mL (±1430) (*p* < 0.01). A significant correlation was found between the level of preoperative NT-proBNP and NYHA classes (r = 0.33, *p* < 0.001), pre-operative LVEF (r = −0.35; *p* < 0.001), and the level of hs-TnT (r = 0.4; *p* < 0.001). Of the 15 patients who received ECMO for cardiogenic shock, 6 were fatal as a result gradually increasing multiple organ dysfunction syndrome. The mean NT-proBNP value in the group of patients requiring ECMO who died was 10,274 pg/mL (±7628) and was significantly higher compared to the patients requiring ECMO support who survived (5056 pg/mL (±3102)) (*p* < 0.05). The total 30-day mortality was 3.7% versus 3.5% (expected mortality was calculated using EuroSCORE II (www.euroscore.org, 30 April 2022). 

## 5. Discussion

The use of ECMO should be considered at an early stage of treatment in a haemodynamically unstable patient after heart valve surgery if that patient has low systolic pressure, low cardiac output, and, as a consequence, insufficient tissue perfusion in which clinical stabilization is not achieved despite the use of conservative treatment in combination with the use of catecholamines [6,22]. The administration of oxygenated blood to the arterial system with appropriate kinetic energy, generated by the ECMO pump, ensures the adequate perfusion of peripheral tissues and relieves the heart muscle by promoting its regeneration [23,24,25]. Therefore, knowledge of the predictors of postoperative cardiogenic shock that do not respond to pharmacological treatment is extremely important because it enables the identification of patients at risk of this complication, thus enabling the early implementation of ECMO, which increases the patient’s chances of survival.

The presented study showed that the level of NT-proBNP determined one day prior to heart valve surgery was an independent predictor of postoperative cardiogenic shock requiring extracorporeal membrane oxygenation, although a significant predictive value in the univariate analyses has also been established for the parameters of the red blood cell system, such as haemoglobin and red cell distribution width. NT-proBNP is a prohormone secreted into the blood by cardiomyocytes (mainly the left ventricle). Due to the fact that the active form of BNP is actively involved in maintaining cardiovascular homeostasis, NT-proBNP is currently widely used in the diagnosis and progression of heart failure [26,27]. 

In the severe valvular heart defects, there is a pressure and/or volume overload of the left ventricle muscle, which leads to an increase in NT-proBNP secretion by cardiomyocytes [11,12,13]. Prolonged left ventricular wall overload is the cause of the progressive myocardial degenerative process involving gradual cardiomyocyte necrosis and fibrosis [28,29]. Very high NT-proBNP values present in the blood serum of patients with hemodynamically significant valvular heart disease may indicate the decompensation of an overloaded left ventricular muscle, which can be confirmed by a significant correlation between the NT-proBNP level and NYHA classes, the pre-operative hs-TnT level, and the LVEF demonstrated in this study.

The results of the present study indicated that patients with high preoperative NT-proBNP values undergoing heart valve surgery may be exposed to postoperative severe cardiogenic shock that is resistant to conservative treatment, which will require the use of advanced mechanical circulatory support techniques. The trend toward statistical significance of the cardiopulmonary bypass time parameter for the primary endpoint demonstrated in this study may also indicate that a decompensated myocardium is particularly vulnerable to the adverse effects of the non-physiological conditions prevailing during extracorporeal circulation. Moreover, the results of this study may also suggest that an earlier qualification of a patient for the surgical treatment of heart valve disease with less-advanced myocardial injury may be associated with an improved prognosis.

## 6. Conclusions

To the best of our knowledge, this is the first report showing the prognostic significance of NT-proBNP measured one day before heart valve surgery in the prediction of cardiogenic shock requiring the use of ECMO in the postoperative period. The results of our research may be helpful in improving the qualifications and perioperative care of patients undergoing heart valve surgery. This was a single-center study that included a limited number of participating patients. In future studies, enlarging the group may allow for confirmation of the results obtained.

## Figures and Tables

**Figure 1 jcm-11-05493-f001:**
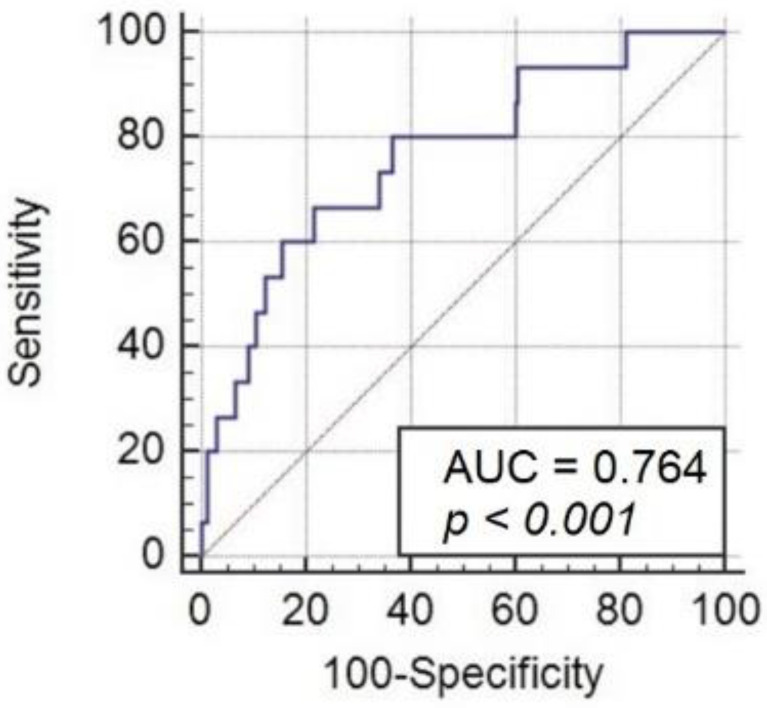
Area under the receiver operating characteristic (AUC) ROC curve of NT-proBNP for cardiogenic shock requiring extracorporeal membrane oxygenation following valve surgery.

**Table 1 jcm-11-05493-t001:** Baseline characteristics of the study population.

Preoperative Characteristics of Patients (*n* = 610)	Values All Patients	Values Patients with ECMO (*n* = 15)	Values Patients without ECMO (*n* = 595)	*p*-Value
Age, years *	63 ± 12	63 ± 11	65 ± 12	Ns
Male: men, *n* (%)	351 (57%)	8 (53%)	343 (57%)	Ns
Body mass index, kg/m^2^ *	28 ± 8	26 ± 6	27 ± 8	Ns
EuroSCORE II, % *	3.5 ± 3.1	3.9 ± 3.5	3.5 ± 3.0	0.04
NYHA, (classes) *	2.5 ± 0.5	3 ± 0.5	2.5 ± 0.6	0.03
LV ejection fraction, % *	57 ± 12	55 ± 12	60 ± 12	0.04
TAPSE, mm *	22 ± 8	21 ± 7	22 ± 7	Ns
RVSP, mmHg *	44 ± 17	48 ± 17	40 ± 16	0.04
Atrial fibrillation, *n* (%)	266 (43%)	6 (40%)	260 (43%)	Ns
Diabetes mellitus, *n* (%)	113 (18%)	3 (20%)	110 (18%)	Ns
NT-proBNP, pg/mL *	2003 ± 1532	7053 ± 3532	1875 ± 1430	0.002
Hs-TnT, ng/L *	34 ± 28	91 ± 58	28 ± 15	0.009
Creatinine, mg/dL *	0.9 ± 0.5	1.4 ± 0.7	0.8 ± 5	0.02
Hemoglobin, g/dL *	13.6 ± 1.5	13.3 ± 1.3	13.8 ± 1.4	0.04
Red cell distribution width, % *	14.2 ± 1.7	15.1 ± 1.7	13.8 ± 1.6	0.009
Intraoperavite and postoperative characteristics of patients:				
AVR, *n* (%)	313 (51%)	7 (46%)	306 (51%)	Ns
AVP, *n* (%)	17 (3%)	Ns	17 (3%)	Ns
MVR, *n* (%)	112 (18%)	3 (20%)	109 (18%)	Ns
MVR + AVR, *n* (%)	53 (9%)	2 (13%)	51 (8%)	0.04
MVP, *n* (%)	115 (19%)	3 (20%)	112 (18%)	Ns
Additional procedureCABG, *n* (%)	90 (14%)	2 (13%)	88 (15%)	Ns
Aortic cross-clamp time, min *	101 ± 32	122 ± 39	98 ± 30	0.01
Cardiopulmonary bypass time, min *	125 ± 55	143 ± 61	118 ± 43	0.03
Postoperative major blending, *n* (%)	47 (8%)	4 (26%)	43 (7%)	0.009

Values are represented by the mean * and a measure of the variation of the internal standard deviation. Abbreviations: AVP: aortic valve plasty, AVR: aortic valve replacement, CABG: coronary artery bypass graft, MVP: mitral valve plasty, MVR: mitral valve replacement, Hs-TnT: high-sensitivity Troponin T, NT-proBNP: n-terminal of the prohormone brain natriuretic peptide, LV: left ventricle, NYHA: New York Heart Association, RVSP: right ventricular systolic pressure, TAPSE: tricuspid annular plane systolic excursion.

**Table 2 jcm-11-05493-t002:** Univariate analysis of the predictive factors for the primary endpoint.

Variable	Odds Ratio	95% CI	*p*-Value
Hemoglobin, g/dL	0.763	0.625–0.967	0.02
NT-proBNP, pg/mL	1.020	1.009–1.036	0.001
RDW, %	1.327	1.083–1.626	0.006
Cardiopulmonary bypass time, min *	1.048	0.986–1.114	0.07

Abbreviations: NT-proBNP: n-terminal of the prohormone brain natriuretic peptide, RDW: red cell distribution width, * denotes the variable that obtained the value closest to achieving statistical significance (*p* < 0.05).

## Data Availability

Research data available from the author of the publication.

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
