# Peer review of "N-Terminal of the Prohormone Brain Natriuretic Peptide Predicts Postoperative Cardiogenic Shock Requiring Extracorporeal Membrane Oxygenation"

_jcm, 2022, doi:10.3390/jcm11195493_

Round 1

Reviewer 1 Report

Interesting and clear. Onyl minor comments:

1) please add the sample size with and without ECMO at the top of the tables

2) were there any difference in outcome and NTproBNP for patients receiveng or not also a CABG or tricuspidal repair?

3) which were the pathophysiological causes of the CS? Did the outcome vary according to this? Were there some cases of huge pericardial effusion or systemic infection?

Author Response

Reviewer 1.

Thank you very much for your interesting review.

The table 1 has been supplemented as recommended.

There were no statistically significant differences in the preoperative NTproBNP level between the subgroups with or without an additional procedure such as CABG. In addition, there was no higher incidence of the primary endpoint in the subgroups with additional CABG (see Table 1).

It seems that one of the main reasons for the occurrence of cardiogenic shock in the postoperative period was the degree of preoperative myocardial damage overlapped by intraoperative circumstances, including the duration of extracorporeal circulation.

Reviewer 2 Report

Dear Author,

Thank you.

Comments:

1. Are you a single author and are you write and search all data?

2. Please clarify the exact definition of post-op cardiogenic shock: LCOS?

3. Please inform in what exact stage did you conduct the ECMO? In operating room, before transfer in ICU? In ICU?Before weaning from CPB or after?

4. As knowing the combined operation (any heart valve surgery+cabg) increase the mortality and morbidity rate. Maybe you had to remove the pts who underwent combined operation?

5. Please inform us, what exact softwave did you implement for analysis?

6.  ''Due to the lack of stabilization of hemodynamic parameters, IABP was replaced with ECMO in 5 cases''. What did you mean? While pts were stabilized you switched the IABP to ECMO?Why/

7. Table 1. Did you include the pts underwent CABG only?90pts?If yes, please remove these pts from analysis.

8. Table 1. Please clarify major bleeding. Post-op?Intra-op?

9. Please avoid self-citation (remove ref 10 and 14)

Thank you

Author Response

Reviewer 2.

Thank you very much for your interesting review.

1.YES, I am the only author of this publication.

I conducted the study myself.

I have developed a project.

I obtained the approval of the institution.

I independently carried out the observation of patients and created data bases.

I conducted the observation and analysis.

I checked the current literature and wrote the manuscript.

2. Postoperative cardiogenic shock described as postcardiotomy shock (PCS) in the available literature remains poorly defined. The prevalence and outcomes of PCS are widely reported, mainly due to the lack of a clear definition. The reported definitions include the inability to wean from extracorporeal circulation (CPB) requiring mechanical circulatory support (MCS), the need for an MCS within 48 hours, or the need to use an MCS at any time during a hospital stay following cardiac surgery. As hinted at, the introduction to the article has been changed.

3. The average duration of ECMO use was approximately 18 hours from the end of surgery. The reason for the use of ECMO was persistent cardiogenic shock with concomitant symptoms of organ hypoperfusion despite the support of pressor amines, the establishment of IABP or the inability to discontinue cardiovascular circulation in the postoperative period.

4. In each case, the main indication for surgery was a significant valvular defect.

As the comparison in Table 1 shows, the CABG procedure in the presented study did not increase the risk of ECMO use.

5. Stistcal analysis was performed with the IBM SPSS soft ware, version 2.0.

6. This sentence indicates the fact that in 8 patients IABP was used due to difficulties in discontinuing cardiopulmonary bypass. In 5 cases, due to the continuing severe cardiogenic shock despite IABP, IABP was replaced with ECMO. Also in the present study, the main endpoint included the most severe cases of cardiogenic shock not reactive to pressor amines and IABP.

7. No. These were patients with significant valve disease who underwent an additional CABG procedure.

8. It was postoperative bleeding.

9. In line with the recommendation, citations 10 and 14 have been deleted.

Round 2

Reviewer 2 Report

Dear Author,

Thank you for your revised manuscript.

Thank you for your answers.